# Advancements in Life Tables Applied to Integrated Pest Management with an Emphasis on Two-Sex Life Tables

**DOI:** 10.3390/insects16030261

**Published:** 2025-03-03

**Authors:** Zhenfu Chen, Yang Luo, Liang Wang, Da Sun, Yikang Wang, Juan Zhou, Bo Luo, Hui Liu, Rong Yan, Lingjun Wang

**Affiliations:** 1Department of Parasitology, Zunyi Medical University, Zunyi 563000, China; m19985468519@163.com (Z.C.); yang18212755200@163.com (Y.L.); m17586533144@163.com (L.W.); xiaokai20240226@163.com (D.S.); wykang1011@163.com (Y.W.); zj412175@163.com (J.Z.); luobozmu@163.com (B.L.); liuhui@zmu.edu.cn (H.L.); 2Laboratory of Evolutionary and Functional Genomics, School of Life Sciences, Chongqing University, Chongqing 401331, China; 3NHC Key Laboratory of Parasite and Vector Biology, National Institute of Parasitic Diseases, Chinese Center for Disease Control and Prevention, Shanghai 200025, China

**Keywords:** age stage, two-sex life table, population dynamics, pest control, application, integrated pest management

## Abstract

Life tables are crucial for unraveling the complex dynamics of pest populations, providing insights into demographic characteristics across different developmental stages and sexes. While numerous studies have highlighted the potential of life table data, such as mortality and emergence rates, in formulating effective IPM strategies, reports of successful implementation in real-world scenarios are scarce. A significant issue is the disconnect between theory and practical application, likely due to a lack of comprehensive research on utilizing the data generated by this tool. This review summarizes the research progress on life tables, particularly their applications in pest control, with the aim of providing a reference for the implementation of effective pest management strategies.

## 1. Introduction

Insect life tables are essential tools in the systematic study of insect populations. These techniques primarily employ parameters such as survival rates, instantaneous mortality rates, mortality distributions, and life expectancies to characterize the life history of insects. Their applications include the fields of insect development, biological control, evolution, and invasion biology. In both ecology and entomology, life tables are utilized to analyze the life cycles and population dynamics of diverse biological populations [1].

Life tables originated in the fields of human sociology and demography, where they were employed to describe and predict changes in population mortality, birth rates, and age distributions [2]. In ecology, population life tables serve a similar purpose, focusing on the dynamics and survival of biological populations. However, traditional life tables have typically focused on a single sex, usually females, which limits their ability to capture the full dynamics of the entire population [3]. Advancements in entomology and the development of new research techniques have led to continuous improvements in classical insect life table methods [4]. The emergence of life tables reflects a deeper exploration of biological traits and population heterogeneity. These life tables analyze the survival rates and reproductive capacities of both sexes across various developmental stages, offering a more comprehensive model of population dynamics. By considering the biological characteristics and behavioral differences between sexes, these tables facilitate a more accurate assessment of the population growth potential and the effectiveness of control strategies [5].

For life tables with raw data on development, the TWOSEX-MSChart program [6] is a versatile tool that can be used to analyze the raw life table data of related insects, providing detailed insights into their development, survival, and daily reproductive capacity. The age-stage-specific survival rate (*s_xj_*: the probability that a newborn egg will survive to age *x* and stage *j*); the age-specific survival rate (*l_x_*: the probability that a newborn egg will survive to age *x*); the female fecundity (F: eggs/female); the age-stage-specific fecundity (*f_xj_*: the number of hatched eggs produced by a female adult at age *x*); the age-specific fecundity (*m_x_*: the number of eggs per individual at age *x*); and the age-specific maternity (*l_x_m_x_*: the product of *l_x_* and *m_x_*) were calculated. All the population parameters, including the intrinsic rate of increase (*r_m_*), were calculated using the Lotka–Euler equation with the age indexed from zero; the finite rate of increase (*λ*) was calculated as *λ* = *er*; the net reproductive rate (*R*_0_) is the sum of all *l_x_m_x_* (age-specific maternity), which considers the survival rate; and the mean generation time (*T*) is the length of time required by a population to increase to *R*_0_-fold of its size as time approaches infinity and the population settles down to a stable age-stage distribution. Key indicators, such as *r_m_*, *λ*, *R*_0_, and *T*, are essential for evaluating the biological characteristics of insect populations [7]. By recording these parameters, we can infer valuable insights into the structure of the insect population, its growth and decline patterns, and other ecological aspects. Life tables serve as an important tool for studying population ecology and for summarizing the survival and reproductive potential of insect populations on different hosts and under various environmental conditions [8].

Insect life tables are pivotal instruments within the realm of insect population ecology and pest management. Their utility extends across a spectrum of research domains, including, but not limited to, pest control, resistance dynamics, extensive insect breeding, the harnessing of biological control agents, and the enhancement of plant resistance [6]. The formulation of a life table encapsulates key statistical metrics of an insect population, thereby offering a structured framework for the documentation and computation of all demographic shifts throughout the insect’s life cycle. A thorough examination of life tables stands as a cornerstone for the management of insects, resistance profiling, the interplay of predators and prey, biological control strategies, and the logistics of large-scale insect cultivation [9].

This article delves into the application of life table methodologies in the realm of integrated pest management, with a pronounced emphasis on biological and chemical control measures. It endeavors to provide robust methods and advanced technologies that enable a scientifically rigorous assessment of pest control efficacy. The ultimate goal is to identify and select the most efficacious control strategies for prospective implementation.

## 2. Application of Life Tables in Biological Control of Pests 

### 2.1. Application in Agricultural Pest Control

Agricultural pests pose a significant threat to crops and agricultural production, impacting biodiversity and causing substantial economic losses. In response to these challenges, modern agriculture increasingly advocates for IPM strategies, which rely heavily on the construction and analysis of pest life table data. These data provide critical insights into how different environmental variables, control strategies, and biological interactions influence pest population growth rates, mortality risks, and reproductive success [10]. Life table technology is indispensable for optimizing pest control measures, ensuring crop safety, and maintaining a healthy ecosystem [11]. By serving as a scientific bridge between theoretical research and practical applications, life tables contribute to the sustainable development of pest management programs (Figure 1 and Appendix A).

#### 2.1.1. Evaluation of the Pest Control Capacity of Natural Enemies of Pests

In agricultural ecosystems, biological control methods that leverage the relationships between organisms and their environment, as well as among different species, are gaining increased attention for protecting ecosystems and ensuring environmental safety [12]. Given that agricultural pests pose a significant threat to global biodiversity and agriculture, natural enemies play a crucial role in managing pest populations [13]. The most traditional method for evaluating the pest control ability of natural enemies is to construct insect life tables. By documenting the growth and development of pests under the pressure of natural enemies, this method assesses the pest control ability of natural enemies from the perspective of pest population dynamics. Natural enemies are usually classified as predators, parasitoids, and pathogens [14].

Predatory natural enemies are potential biological control agents for rapidly reproducing pests, such as mites [15], whiteflies [16], and thrips [17]. The intrinsic natural growth rate *r_m_* of a population is of great interest as a key parameter in entomology and is considered particularly important in the study of predators [18]. Among the common predatory natural enemies, Coccinellidae stands out due to being known for preying on aphids, scale insects, whiteflies, and spider mites [19]. The Chrysopidae, commonly known as lacewings, are renowned for their formidable predatory capabilities against a broad spectrum of agricultural and forestry pests, occupying a crucial position in biological control strategies [20]. Both the larval and adult stages of these insects actively prey upon a diverse range of pests, such as aphids, whiteflies, mites, and cotton bollworms. Moreover, Pentatomidae, which includes predatory stink bugs, contributes significantly to natural pest management by feeding on a diverse range of hosts, such as moth larvae, various beetles, and leaf beetles [21].

By utilizing a two-sex life table for analysis, the feeding potential and efficacy of *Orius strigicollis* Poppius (Heteroptera: Anthocoridae) were quantified under the selection pressures of different prey densities [22]. It was found that, when feeding on 10 *Pectinophora gossypiella* Saunders (Lepidoptera: Gelechiidae) eggs, the females showed significantly increased feeding capabilities and the highest *R*_0_ and gross reproductive rates (*GRR*) were achieved; in addition, *O. strigicollis* showed a higher and similar *r* (faster development and highest survival rates) when fed with 10 or 15 eggs of *P. gossypiella*, owing to a higher fecundity and shorter or faster development times. This revealed the potential of *O. strigicollis* as an effective predator of *P. gossypiella* [22]. By combining the Holling disc equation with the age-stage, two-sex life table technique, it was demonstrated that each life stage of *O. strigicollis* exhibited a type II functional response when presented with third-instar nymphs of *Bemisia tabaci* Gennadius (Hemiptera: Aleyrodidae) and *Trialeurodes vaporariorum* Westwood (Hemiptera: Aleyrodidae) [23]. The calculated prey handling time for *O. strigicollis* was shorter when fed *T. vaporariorum;* additionally, the nymphal development and the total pre-oviposition period of adult females of *O. strigicollis* were significantly shorter when fed *B. tabaci*, confirming its potential application as a biocontrol agent in integrated pest management, especially for controlling *B. tabaci* efficiently [23]. The consumption, development, and reproduction of three native Chinese phytase enzymes—*Neoseiulus californicus* McGregor (Mesostigmata: Phytoseiidae), *Neoseiulus barkeri* Hughes (Mesostigmata: Phytoseiidae), and *Amblyseius orientalis* Ehara (Mesostigmata: Phytoseiidae)—were comprehensively evaluated using life table parameters, with *Amblyseius swirskii* Athias-Henriot (Mesostigmata: Phytoseiidae) serving as a control. When fed *Polyphagotarsonemus latus* Banks (Acari: Tarsonemidae), *A. orientalis* exhibited the highest consumption rate, the shortest development time, and the highest cumulative fecundity. Notably, its *r_m_* was 0.12, and *A. orientalis* was the sole species among the three that was anticipated to experience population growth when fed *P. latus*. It is recommended that *A. orientalis* be considered as a potential biological control agent for this pest [24].

Parasitic natural enemies are also crucial in biological control. By examining the morphological features, parasitism rates, and developmental history of key parasitic natural enemies, valuable insights can be gained. These insights are vital for conserving the biodiversity of these natural enemies and utilizing local species for pest control [25]. In the realm of ecology, the parasitic wasp families Trichogrammatidae [26], Ichneumonidae [27], Braconidae [28], and Pteromalidae [29] have emerged as crucial biocontrol agents. They are proficient at managing the populations of harmful insects by strategically employing parasitic tactics. Through the analysis of two-sex life tables, it was found that *Sitotroga cerealella* Olivier (Lepidoptera: Gelechiidae) reared on maize exhibited significantly higher values of *λ*, *r*, and *R*_0_. Additionally, the mean parasitism, mean adult emergence, longevity of adults, and total adult longevity of *Trichogramma chilonis* Ishii (Hymenoptera: Trichogrammatidae) were recorded as being the highest on *S. cerealella* eggs reared on maize. According to evolutionary models, more females will oviposit in large hosts than in small hosts, which is consistent with the finding that maize, being a larger host, supports a higher proportion of female offspring in *T. chilonis.* Identifying the most susceptible and favorite host (maize) will help us improve the large-scale production of *T. chilonis* under laboratory conditions [30]. Temperature significantly influenced the biological parameters of *Trichogramma euproctidis* Girault (Hymenoptera: Trichogrammatidae); at 32.5 °C, it achieved peak parasitism rates on *Ephestia kuehniella* Zeller (Lepidoptera: Pyralidae) eggs, produced the highest number of female offspring, and attained the greatest survival rate [31]. This strain of *T. euproctidis* is adapted to high temperatures and harsh environmental conditions, showing potential for use in integrated management programs in Southwest Iran [31]. The fertility life table parameters of *Tetrastichus howardi* Olliff (Hymenoptera: Eulophidae) parasitizing *Plutella xylostella* Linnaeus (Lepidoptera: Plutellidae) and studies on the effects of the natal host on the behavior of *T. howardi* towards host volatiles and the parasitism rate have shown that the *Ro* and *r_m_* of *T. howardi* are 13.6 and 0.124, respectively, while the mean generation time is 20.9 days. Furthermore, the natal host *Tenebrio molitor* Linnaeus (Coleoptera: Tenebrionidae) or *P. xylostella* does not affect the fitness or parasitism rate of *T. howardi*. Therefore, *T. howardi* reared on the artificial host maintains its attraction to and potential to parasitize *P. xylostella* [32].

As the demand for sustainable agricultural development grows, insect pathogens are gaining increasing attention for pest control [33]. Nonetheless, systematic research methods are required to evaluate their efficacy and optimize strategies. The age-stage, two-sex life table technique is a powerful tool that can comprehensively assess pest survival, reproduction, and population growth and quantify the impacts of insect pathogens on these parameters [34]. Constructing these life tables allows for the precise evaluation of the long-term effects of insect pathogens and the refinement of field strategies. Using an age-stage, two-sex life table technique to study the role of insect pathogens in pest control can provide a robust scientific basis for biological control and integrated pest management, promoting sustainable agricultural development. The impacts of two entomopathogenic fungi, *Beauveria bassiana* Vuill (Hypocreales: Cordycipitaceae) and *Metarhizium anisopliae* Sorokin (Hypocreales: Clavicipitaceae), on the life history parameters of *Coccinella septempunctata* Linnaeus (Coleoptera: Coccinellidae), a significant generalist predator, were examined. The findings indicated that entomopathogenic fungi do not have any significant side effects on the performance or biology of *C. septempunctata*, revealing the nonpersistent impact of biocontrol agents on pest control [35]. An investigation into the biological and biochemical impacts of *M*. *anisopliae*, *B*. *bassiana*, and *Purpureocillium lilacinum* Thom (Hypocreales: Ophiocordycipitaceae) on the third-instar larvae of *Culex pipiens* Linnaeus (Diptera: Culicidae) laboratory colony revealed that *M. anisopliae* demonstrated a superior efficacy. It displayed the highest larval mortality (88%) and the shortest LT_50_ (22.6 h), indicating its potential as the most effective biological control agent among the fungi assessed. In addition, the study revealed a reduction in female fecundity, the number of hatched eggs, the pupation percentage, and the adult emergence percentage, as well as changes in biochemical indicators. Thus, *M. anisopliae* has proven to be an effective biological control agent for *C. pipiens* [36]. Exposure to sublethal (LC_20_) and lethal concentrations (LC_50_) of *B*. *bassiana* significantly impacted the parental generation (F_0_) of *Spodoptera exigua* Hubner (Lepidoptera: Noctuidae), and these effects cascaded to the demographic parameters of the first filial generation (F_1_). The infected F_1_ offspring exhibited a decreased *r*, an extended *T*, and a reduced *R*_0_. Furthermore, the fecundity of the *B. bassiana*-infected groups was notably lower than that of the control. These findings highlight the enduring effects of *B. bassiana* on the biological parameters and population dynamics of *S. exigua*, underscoring its potential as an eco-friendly biopesticide for pest management [37].

#### 2.1.2. Evaluation of the Pest Control Capacity of Bio-Pesticides

Pest outbreaks and resilience pose significant threats to food security, highlighting the need for effective pest control measures. Traditional chemical pesticides have limitations, including environmental pollution and the development of pest resistance. Therefore, new pest control technologies, such as biopesticides, are essential to address these challenges and provide sustainable solutions for crop protection [38]. Biopesticides, including microbial biopesticides and other biocontrol agents, offer a promising alternative to traditional chemical pesticides. These biologics are derived from various sources, such as microorganisms (e.g., metabolites) and plants (e.g., secretions; essential oils; and extracts of bark, roots, and leaves) [39]. They encompass a diverse array of microbial pesticides [40], including bacteria, fungi, and viruses, as well as biochemical pesticides [41], such as pheromones and natural plant inducers, and plant-derived pesticides [42], such as plant extracts. These components leverage the natural antagonistic properties of these organisms and compounds to disrupt pest life cycles, deter feeding, or induce resistance in host plants, thereby providing sustainable solutions for agriculture and environmental protection [40,41,42]. To better understand the factors and effects of biopesticides on pest control, it is essential to evaluate their long-term impacts on pest population sizes and dynamics, as well as their potential effects on pests and other beneficial insects. Life table evaluation techniques can be employed to assess these impacts, providing valuable insights for the effective use of biopesticides in IPM strategies [43]. This evaluation will help optimize the application of biopesticides and enhance their role in sustainable agriculture. Research has shown that treating second-instar larvae of *Helicoverpa armigera* Hubner (Lepidoptera: Noctuidae) with HearNPV, a nucleopolyhedrovirus to *H. armigera*, and then parasitizing them with *Habrobracon hebetor* Say (Hymneoptera: Braconidae) results in sublethal effects on *H*. *hebetor*, including reduced longevity and fecundity at sublethal concentrations (e.g., LC_30_) and decreased population growth parameters (e.g., *R*_0_ and *r_m_*). However, under field conditions, *H*. *hebetor* can still effectively control *H*. *armigera* when released 2 days after HearNPV application, suggesting that their combined use in pest management is feasible [44].

#### 2.1.3. Screening of Insect-Resistant Plant Species

Host resistance plays a vital role in integrated pest management. Host plant resistance is a key component of pest management and one of the most appreciated control strategies in advanced agriculture [45]. This phenomenon is the result of heritable plant traits that cause plants to suffer less damage compared to plants lacking these qualities. Insect-resistant crop varieties, such as rice, maize, and cotton [46,47], reduce pest populations by enhancing their tolerance to insect damage. Three types of resistance determine the relationship between insects and plants: antibiosis, antixenosis, and tolerance [48]. In recent years, the field of biotechnology has witnessed remarkable advancements, including gene-editing technologies, the interspecies transfer of resistance genes, and the enhancement of systemic acquired resistance. These innovations hold great promise for the development of pest-resistant crops. Life tables have emerged as a powerful tool to effectively evaluate the potential of these novel biotechnologies, by integrating pest severity data across diverse plant species, combining the extent of pest infestations on different plants, comparing the degree of variation in damage between varieties, and providing a qualitative evaluation of plant resistance to pests, particularly the screening of resistant and susceptible varieties. For example, the biological parameters and fecundity life table of *Melanaphis sorghi* Theobald (Hemiptera: Aphididae), a pest that infests sorghum crops, were estimated on 15 sorghum hybrids. The results identified specific sorghum varieties that were less suitable for the pest and showed resistance to *M*. *sorghi* [49]. The *r*, *R*_0_, and *T* of *Toxoptera aurantii* Boyer de Fonscolombe (Hemiptera: Aphididae) on six different tea tree varieties were analyzed to determine their population dynamics and host adaptation, which were used to guide integrated pest management and the screening of *T. aurantii*-resistant host varieties [50].

**Figure 1 insects-16-00261-f001:**
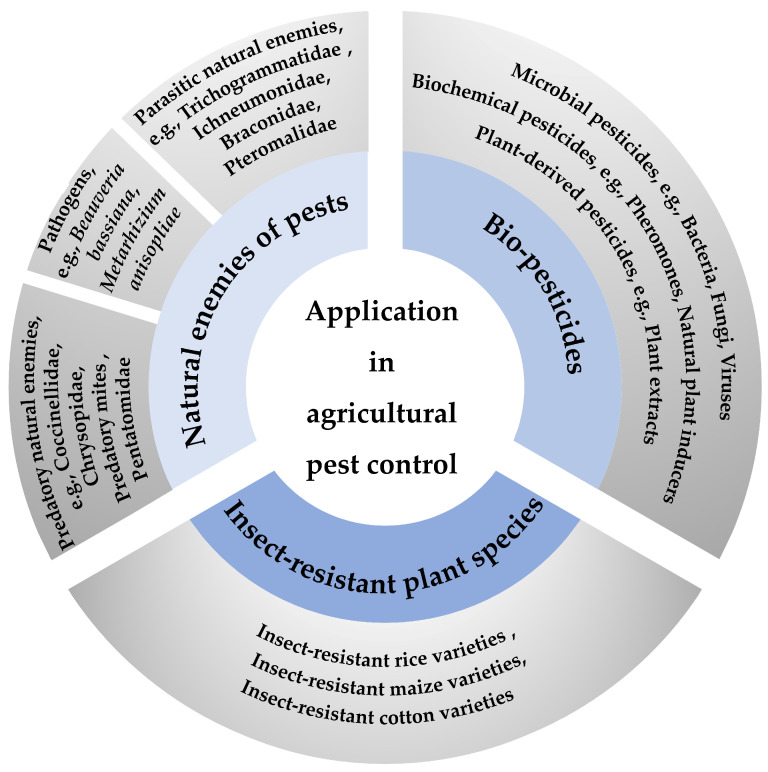
Summary of application of life table in agricultural pest control. Three primary strategies—natural enemies of pests, biopesticides, and insect-resistant plant species—were used for agricultural pest control. Natural enemies of pests: predatory natural enemies [19,20,21], pathogens [35,36], and parasitic natural enemies [26,27,28,29]. Biopesticides: microbial [40], biochemical [41], and plant-derived pesticides [42]. Insect-resistant plant species: rice [46], maize, and cotton [47]. Specific examples are provided in the outer circle.

### 2.2. Application in Vector Insect Control

The effective management of vector insects is of paramount importance in curbing the dissemination of infectious diseases. Vector insects serve as critical conduits for the transmission of pathogens to both human and animal hosts, thus occupying a central position in public health and disease prevention initiatives [51]. Numerous infectious diseases rely heavily on vector insects for their transmission. For instance, mosquitoes, which are among the most prevalent vectors, are primarily responsible for the propagation of illnesses such as malaria, dengue fever, yellow fever, and Zika virus disease. Their broad distribution and biting behaviors significantly contribute to their role as vectors [52]. In Diptera, certain species have been shown to transmit intestinal diseases such as cholera and typhoid fever [53]. These insects facilitate pathogen transmission through direct contact with the hosts, thereby necessitating robust control measures to prevent disease outbreaks. Life tables offer a valuable tool for understanding the life cycle and population dynamics of vector insects. By providing structured and data-driven insights, life tables enable the formulation of targeted and effective control strategies. These approaches can be customized to target the unique traits and weaknesses of vector insect populations, thus optimizing the effectiveness of disease control initiatives. Evaluating the life history characteristics of vector insects using life tables can help us understand their survival rate, longevity, and fecundity at different stages; this approach aids in the development of a better understanding of the population dynamics of vector insects. Additionally, life table parameters can be used to predict the population growth rate and trends in the changes in vectors. Ultimately, this information can be used to formulate targeted strategies, providing a scientific basis for the implementation of effective vector control measures [54] (Figure 2 and Appendix A).

#### 2.2.1. Assessing the Risk of Transmission by Vector Insects

Life tables hold crucial significance in assessing the transmission risk associated with vector insects. They serve as an indispensable tool, providing essential indicators for evaluating the transmission potential of these vector pests and the corresponding disease transmission risk. By analyzing life table parameters, such as the longevity, survival rates, and reproduction rates, we can gain a comprehensive understanding of the population growth dynamics and disease transmission capacity of the insects under various environmental conditions. With these insights, we can then develop targeted interventions aimed at reducing the population size and mitigating the disease transmission risk posed by vector pests. By analyzing the life table of *Anopheles balabacensis* Baisas (Diptera: Culicidae), the main vector of *Plasmodium knowlesi* Sinton and Mulligan (Haemosporida: Plasmodiidae), to assess the potential for the nonzoonotic transmission of *P. knowlesi*, it is possible to estimate the survival rate of mosquitoes better. These estimates of mosquito survival rates enable an assessment of the duration and likelihood of parasite development in a mosquito and its transmission to give rise to a secondary case [55]. By studying the population growth and survival rates of the malaria vector *Anopheles stephensi* Liston (Diptera: Culicidae) in different water pollutants and analyzing the impact of changes in water quality on this particular mosquito, the risk of transmission of vector-borne diseases can be assessed, which is an important guide for the development of effective public health strategies and policies [56].

#### 2.2.2. Modeling the Dynamics of Vector Insects

Accurately modeling the population dynamics of vector insects is essential for understanding and controlling the spread of the diseases they transmit. Life table data form the cornerstone of such modeling efforts. By integrating environmental factors, climatic data, and disease transmission parameters, life table data can be used to develop mathematical models that simulate and predict the population changes and disease transmission dynamics of vector pests [57]. These models incorporate various factors, such as environmental conditions, climatic variables, and disease-transmission parameters [57]. The information generated by these models is extremely valuable for guiding the development and refinement of vector-pest-control strategies [58]. By leveraging this information, we can design more targeted and effective interventions to combat the threat posed by vector insects and the diseases they carry. By employing a pseudostage-structured population dynamics model, environmental dependencies were deduced from life cycle observations for *Culex quinquefasciatus* Say (Diptera: Culicidae) and *C. pipiens*. Photoperiodicity and temperature emerged as critical factors influencing the duration of the larval stage. It was discovered that meticulously timed life history observations under natural field settings can accurately predict insect development across the annual cycle [59]. A temperature-dependent phenology model for the whitefly vector has also been developed using the Insect Life Cycle Modeling (ILCYM) software [60]. The impact of temperature on the whitefly’s virus transmission efficiency was assessed via controlled lab experiments at eight constant temperatures (10–25 °C). The vector’s transmission capacity was the strongest at 15 °C (about 70% infection probability) but dropped sharply to <10% at 10 and 20 °C. A nonlinear function describing the temperature-dependent transmission probability of a single adult whitefly was validated using transmission frequencies under fluctuating temperatures. This function, along with life table parameters from the temperature-dependent phenology model, formed a comprehensive temperature-responsive model for predicting PYVV’s spread potential and transmission probabilities. The best-performing risk index was used to create risk maps. These maps not only accurately reflected the virus’s actual occurrence but also predicted high-risk areas where it had not been reported before. Surveillance in western Panama, a predicted high-risk area, led to the virus’s identification there, where it was previously unknown [60].

#### 2.2.3. Interference with the Life Cycle of Vector Insects

The application of life table data is crucial for understanding the life cycle of vector insects and pinpointing key stages for intervention. These data reveal the various developmental stages of vector pests and their associated vulnerabilities. By targeting these stages, such as disrupting reproduction, larval hatching, or adult longevity, we can effectively reduce the population size and disease transmissibility of vector pests. This life-table-based approach provides a strategic framework for developing interventions that disrupt the life cycle of vector insects and mitigate the risk of disease transmission [61]. The entomopathogenic bacterium *Chromobacterium anophelis* Frankland (Flavobacteriales: Flavobacteriaceae) sp. nov. IRSSSOUMB001 was utilized to assess its pathogenic potential against the larval stage of *Anopheles coluzzii* Coetzee and Wilkerson (Diptera: Culicidae), as well as its influence on the reproductive capabilities and transgenerational consequences of mosquitoes. This research confirmed the efficacy of the bacterium in infecting *A. coluzzii*, which serves as a vector for malaria transmission. Furthermore, this study highlighted the pronounced virulence of the strain IRSSSOUMB001 against insecticide-resistant *A. coluzzii* larvae and its capacity to decrease mosquito fecundity and the fitness of its progeny [62].

**Figure 2 insects-16-00261-f002:**
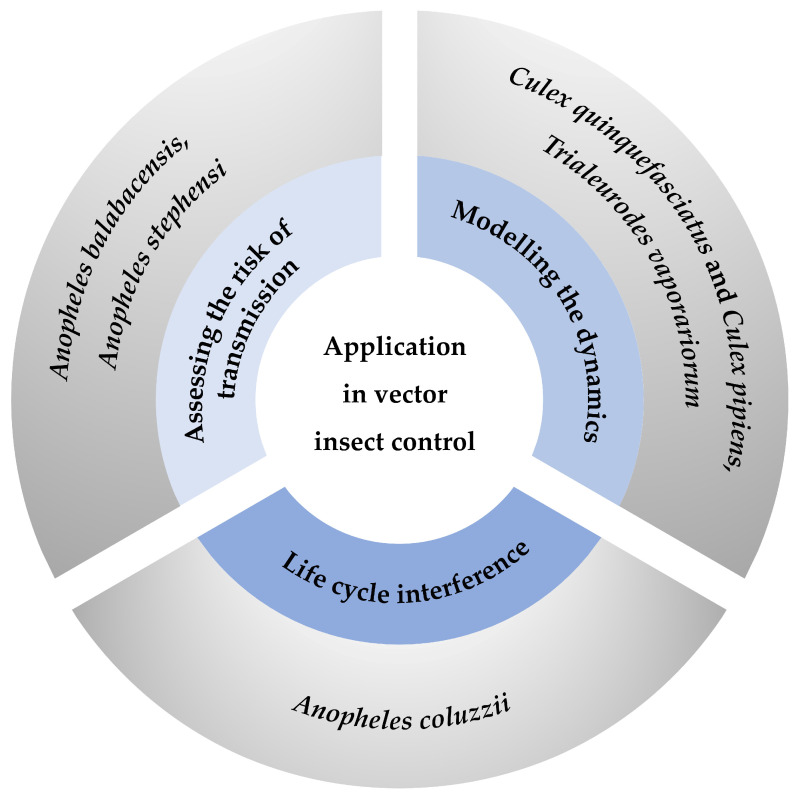
Summary of application of life tables in vector insect control. Three primary strategies for controlling vector insects include assessing the risk of transmission [55,56], modeling the dynamics [59,60], and performing life cycle interference [62]. Each strategy is represented by a segment of the circle, with specific examples provided in the outer circle.

### 2.3. Application in Invasive Pest Control

Biological invasions are increasingly recognized as important spatial processes that drive global change and threaten biodiversity, regional economies, and ecosystem function [63]. The application of life tables to invasive pests is an important tool for the study and management of invasive pest population dynamics, life cycles, reproductive potential, and impacts on ecosystems. Life table analyses can help monitor the population sizes, growth rates, and trends of invasive pests. By collecting data on the life cycle, survival rates, and reproduction rates of the population, life table models can be constructed to predict future population sizes. Life tables provide important information about the life cycle characteristics of invasive pests, such as the hatching rate, development time, sex ratio, and longevity [64]. These data can be used to understand the life history and life history parameters of a pest.

Investigations have employed life table analyses to evaluate the suitability of three plant species—tomatoes, potatoes, and eggplants—as hosts for the invasive mealybug, *Phenacoccus solenopsis* Tinsley (Hemiptera: Pseudococcidae). The results revealed that all the tested host plants were indeed suitable for *P. solenopsis*. Notably, the eggplant host plant exhibited the highest fecundity, *R*_0_, and λ, as well as the longest adult longevity (males: 6.50 ± 0.34 days; females: 24.15 ± 0.50 days). These findings provide critical insights that will significantly aid in the development of targeted and effective management strategies for successfully controlling this invasive pest in major Mediterranean crop systems [65]. The study constructed a life table for *Acanthococcus lagerstroemiae* Kuwana (Hemiptera: Eriococcidae) to analyze the impact of plant nutrient conditions on its population. The results indicated that, under nutrient-deficient conditions (cultivated with water only), *A. lagerstroemiae* had a higher intrinsic rate of increase, finite rate of increase, and net reproductive rate. In contrast, under nutrient-rich conditions (0.1 MS nutrient solution), the mean generation time of *A. lagerstroemiae* was longer. This suggests that *A. lagerstroemiae* performs better on plants under nutrient-deficient conditions. These findings provide a basis for developing environmentally friendly pest management strategies [66].

## 3. Application of Life Tables in Chemical Control of Pests

Chemical application is considered one of the most critical methods of pest control, especially in intensive agricultural practices [67]. This is because the effectiveness of chemical control is significantly influenced by the morphological and developmental stage structure of pest populations, which is crucial for determining the susceptibility of pests to insecticides and ensuring the success of pest management strategies [68] (Figure 3 and Appendix A).

### 3.1. Evaluating the Fitness Costs of Pesticide Resistance in Pests

The frequent use of insecticides has led to the development of resistance in many insects [69]. As pest resistance to insecticides gradually increases, the efficacy of these chemicals diminishes, and in some cases, they may even fail to control the pests, which in turn impacts agricultural production. Therefore, evaluating the fitness costs associated with pesticide resistance in pests is crucial for understanding the dynamics of resistance and developing effective control measures. An investigation into the correlation between imidacloprid resistance and the fitness of the melon aphid *Aphis gossypii* Glover (Hemiptera: Aphididae) was undertaken, offering valuable insights into the potential fitness trade-offs related to resistance against this neonicotinoid insecticide. The imidacloprid-resistant lines (ImR) presented prolonged developmental stages, shortened longevity, and decreased fecundity. Key demographic parameters were significantly reduced in ImR, indicating that resistance has a fitness cost. At the molecular level, the expression of genes related to development and reproduction changed, and some of these genes were downregulated. These findings have important implications for understanding the development and spread of resistance, and they provide a scientific basis for field management, helping to delay the development and spread of resistance [70]. A comprehensive study on the meadow nightshade moth *Spodoptera frugiperda* Smith and Abbot (Lepidoptera: Noctuidae) and its ability to adapt to chlorpyrifos on various host plants has been conducted. The life table parameters of the moths on these different hosts indicated the presence of a fitness cost associated with resistance to chlorpyrifos at both the individual and population levels. This finding suggests that the withdrawal of selective agents from the environment could result in a decrease in resistance levels, presenting an opportunity to restore susceptibility to these agents [71].

### 3.2. Guiding the Selection of Insecticides

Synthetic insecticides have become an integral component of global plant protection strategies [72]. Given their widespread use, selecting the appropriate insecticide has emerged as a pivotal step in effective pest management. This decision-making process necessitates a thorough understanding of both the target pest and its environmental context [73]. Life tables, which offer detailed insights into the life cycle and population dynamics of pests, are instrumental in guiding the selection of insecticides. The choice of insecticide is a complex and multifaceted decision that demands careful consideration of the pest’s biology and the broader ecological implications. Life tables provide essential data on pest life cycles and population dynamics, enabling more informed and targeted insecticide choices. By integrating these data with an understanding of insecticide mechanisms and types, pest management strategies can be optimized to achieve effective control while minimizing their environmental impact. Using an age-stage life table methodology, the sublethal concentration (LC_50_) of triflumizole-pyrimidines against *Laodelphax striatellus* Fallén (Hemiptera: Delphacidae) was evaluated. Compared with those of the F_0_ generation, the *r*, *λ*, and *R*_0_ of the F_5_ generation were significantly lower, suggesting that the LC_50_ of triflumezopyrim may impede the generational growth and reproduction of *L. striatellus*. This evaluation is intended to provide a foundation for future research attempting to elucidate the adaptability and resistance mechanisms of *L. striatellus* in response to sublethal doses of triflumizole pyrimidine [74]. A thorough assessment of alternative pesticide efficacy was carried out by testing the susceptibility of *Bradysia odoriphaga* Yang and Zhang (Diptera: Sciaridae) and *Bradysia difformis* Johannsen (Diptera: Sciaridae) to multiple insecticides. The toxicity of eight pesticides was evaluated against several parasitic species, with a focus on how sublethal doses of dinotefuran and lufenuron affected their life history traits and detoxification enzyme functions. The results indicated that dinotefuran and lufenuron were particularly toxic to *B. odoriphaga* and *B. difformis* out of the tested compounds. Moreover, sublethal concentrations of these pesticides significantly impaired the life history parameters across both generations of the species. These findings offer valuable insights for targeted pest control, underscoring the efficacy of dinotefuran and lufenuron as effective management tools [75].

### 3.3. Guiding the Application of Insecticides

The rational application of insecticides is essential for effective pest control while minimizing the environmental impact. Life table data play a pivotal role in guiding the timing and dosage of insecticide applications, ensuring that interventions are both effective and efficient. A simulation using a sex-based life table can predict the optimal period for pest control and accurately determine the optimal timing and number of chemical control applications [76]. Following the application of pesticides, the population size and age structure of pests undergo significant changes. By analyzing the population structure and considering factors such as the lethality of insecticides for different insect states and age classes, the persistence period of insecticides, and the economic threshold (ET), we can utilize the amphoteric life table and the TIMING—MSChart to predict the effects of insecticide use on the growth and reproduction of pests and their progeny [77]. This approach enables us to simulate the appropriate timing of control measures, thereby optimizing the use of insecticides and enhancing the effectiveness of pest management strategies. A detailed life table was constructed for the coffee berry borer *Hypothenemus hampei* Ferrari (Coleoptera: Curculionidae) by meticulously calculating its survival and reproduction rates across various developmental stages. This comprehensive analysis allowed for the determination of *R*_0_ for each population, thereby pinpointing the optimal timing for the most effective pest control interventions [78]. A comprehensive study was conducted to elucidate the life cycle characteristics of the brown marmorated stink bug *Halyomorpha halys* Stål (Hemiptera: Pentatomidae) under various temperature conditions. Through a comprehensive approach involving both laboratory and field experiments, this study examined the influence of temperature fluctuations on the diverse life stages of *H. halys* in diverse geographical regions within the United States. Furthermore, it quantified the varying patterns of these pivotal parameters at distinct temperature levels, thereby providing a solid scientific foundation for predicting and managing the population dynamics of this insect pest [79].

**Figure 3 insects-16-00261-f003:**
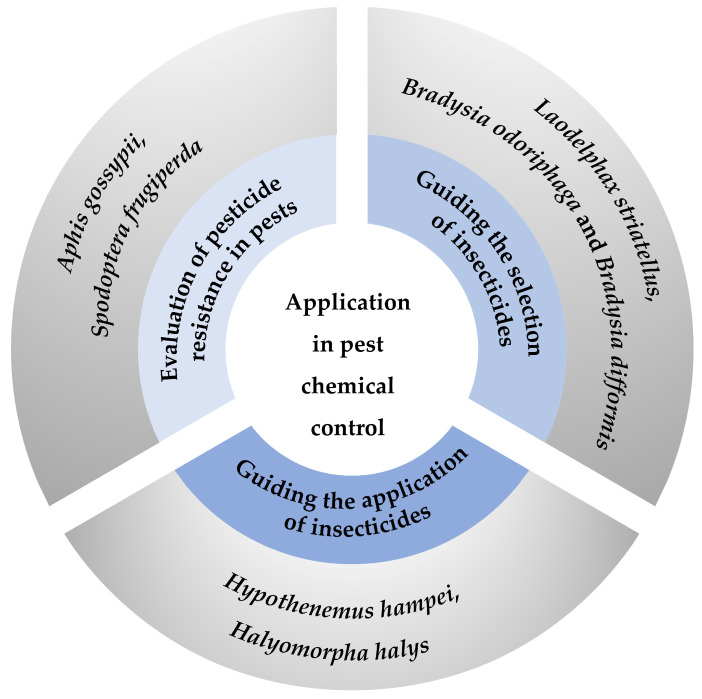
Summary of application of life tables in pest chemical control. Three primary strategies for pest chemical control include evaluating pesticide resistance [70,71] and guiding both the selection [74,75] and the application [78,79] of insecticides. Each strategy is represented by a segment of the circle, with specific examples provided in the outer circle.

## 4. Discussion and Future Perspectives

Life tables have significantly advanced pest management by quantifying critical demographic parameters such as survival rates, developmental timelines, and reproductive output. Studies on predators such as *O. strigicollis* and parasitoids such as *Trichogramma* species have demonstrated their efficacy under controlled conditions, with peak performance linked to intermediate prey densities and host-specific adaptations [22,23,30,80]. Similarly, entomopathogenic fungi (EPFs) show compatibility with predators such as *C. septempunctata*, suggesting potential synergies in multi-agent biocontrol systems [35]. However, these findings remain largely confined to laboratory settings, and the field-scale validation of ecological interactions—such as cascading trophic effects or multi-predator dynamics—is still limited. This gap underscores the need to reconcile controlled-environment insights with the complexity of natural agroecosystems, where biotic factors (e.g., natural enemies) and abiotic stressors (e.g., temperature fluctuations) interact unpredictably [36,71].

A critical limitation of current life table applications lies in their narrow focus on single-species or single-factor interactions. For instance, while fitness costs associated with pesticide resistance—such as a reduced fecundity in imidacloprid-resistant *A. gossypii* [70] or chlorpyrifos-resistant *S. frugiperda* [71]—highlight trade-offs exploitable for resistance management, most studies have neglected population-level evolutionary drivers such as gene flow and selection pressures [81]. Similarly, invasive pests such as *S. frugiperda* exhibit diet-dependent population growth in artificial rearing systems, yet these outcomes often diverge from field realities, where the host plant quality and climate variability modulate invasion success [66,71]. Cross-species comparisons and multi-trophic analyses (e.g., pest–plant–microbe interactions) remain understudied, limiting the generalizability of life table-derived strategies.

Emerging technologies offer promising solutions to scalability and contextual limitations. Automated video tracking, remote sensing, and machine learning algorithms enable the real-time, large-scale monitoring of pest behavior and population trends [62,82], while genomic tools elucidate resistance mechanisms in pests such as *A. gossypii* [70]. Climate-driven models, such as temperature-sensitive phenological frameworks for *T. vaporariorum* [83], integrate environmental variables to predict pest outbreaks with increasing precision. However, the effectiveness of these tools hinges on their validation across diverse agroecosystems and their integration with traditional life table data. Bridging this gap requires adaptive “smart IPM” systems that dynamically synthesize laboratory-derived parameters, field observations, and environmental datasets to optimize intervention timing and spatial targeting.

The context-specific nature of life table outcomes further complicates broad applications. For example, the efficacy of *M. anisopliae* against *C. pipiens* larvae varies markedly with the instar stage and microhabitat conditions [36], while the fitness costs in resistant pests often depend on the host plant chemistry or regional climate patterns [70,71]. Such variability necessitates localized validation and regionally tailored management strategies. Longitudinal studies across multiple generations and agroecological zones are critical to deciphering resistance evolution, predator–prey coadaptation, and the long-term stability of biocontrol agents. Additionally, expanding life table frameworks to assess indirect ecological impacts—such as the effects of insect-resistant crops on pollinators or soil microbiota—would enhance the sustainability of IPM programs [54]. To fully realize the potential of life tables in IPM, future research must prioritize cross-disciplinary integration. Combining population models with food web analytics, genomic insights, and climate resilience frameworks will enable a holistic understanding of pest dynamics. Concurrently, establishing open-access databases of regionally calibrated life table parameters could guide context-specific interventions. By addressing these challenges, life table analyses can transition from a descriptive tool to a predictive engine for designing adaptive, ecologically informed pest management strategies that balance efficacy with environmental stewardship.

## Data Availability

Data sharing is not applicable.

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
