# Peer review of "Advancements in Life Tables Applied to Integrated Pest Management with an Emphasis on Two-Sex Life Tables"

_insects, 2025, doi:10.3390/insects16030261_

Round 1
Reviewer 1 Report
Comments and Suggestions for Authors
A brief summary
Two-sex life table is a beneficial tool that can be applied in support of insect pest management (IPM). The authors of this manuscript conducted extensive literature reviews on this topic with the hope to integrate two-sex life tables in different technologies as part of IPM. The manuscript is organized fairly well. The figures helped visualizing what the authors attempted to convey. Some language improvements are necessary as many areas within the manuscript repetitively used the same words or phrases over and over. These words or phrases in some cased do not even needed to be in the sentence(s). I will try to point out below but ultimately the authors need to go through the entire document to look for things to be improved.
Specific comments:
Some important tips:
- 1. ‘Meticulously examined’ phrase was used numerous times throughout the manuscript. Reword. In many cases the word meticulous is not necessary. When one conducted research, it was automatically a meticulous work that had to be done.
- 2. Spell out species name for the first time then abbreviate throughout document. For example, in line 181 you spelled out Metarhizium anisopliae. In line 187, you can just abbreviate/shorten as M. anisopliae.
- 3. Change manuscript title: I checked a few references that were cited and they did not use two-sex life tables for the studies; the studies only referenced to females. I suggest ‘two-sex life table’ be changed to ‘life table’. Therefore, the manuscript title needs to be changed.
- 4. Where possible, replace the word ‘corn’ with ‘maize’ which the more preferred term in scientific publication.
- 5. Be sure use past tense as appropriate.
Line 101 (page 3)
Put comma after 'exploration' and after 'application'
Lines 111-113 (page 3)
In general and common entomological knowledge, natural enemies consist of predators, parasitoids, and pathogens. Why are pathogens excluded??
Lines 148-149 (page 4)
I am not sure what the authors meant by ‘ecological theater’. This is uncommon, poor choice of phrase. Please rephrase.
Lines 159 (page 4)
Delete duplicate Ephestia kuehniella. Also it is best to reword this way: …, a parasitoid Ephestia kuehniella Zeller egg …
Line 178 (page 4)
Repetitive words, please delete ‘longer term or’.
Line 183 (page 4)
Meticulously is unnecessary à delete.
Lines 186-188 (page 4)
Remove ‘entomopathogenic fungi’. It is a repetitive word. See comments above on species name (under “some important tips”).
Line 199 (page 5)
Improper use of words; change ‘than do’ with ‘compared to’.
Lines 200-203 (page 5)
Again, there is redundancy here – please rephrase.
Line 204 (page 5)
Biotic resistance is incorrect type of resistance. Three type of resistance are antibiosis, antixenosis, and tolerance. Check references.
Line 205 (page 5)
… by these new biotechnological tools … à This is unclear. Which biotechnological tools? The authors have not given introduction about biotechnological tools and all of the sudden talk about this and perhaps this needs to be covered in the introduction section.
Figure 1 (page 5)
In each component use e.g. and remove etc.
For example,
Parasitic natural enemies
e.g. Trichogrammatidae …
Figure 1 (page 5)
Pathogens should be part of natural enemies. Biological agents is not a proper term: predators and parasitoids (parasitic natural enemies) are also biological agents.
Line 224 (page 6)
Change ‘within the fly category (flies)’ to ‘In Diptera’.
Line 260 (page 6)
Delete ‘of two Culex species, namely’. This is unnecessary.
Lines 323-325 (page 8)
How? This is unclear.
Line 329 (page 8)
Age (or instar) … à Not always instar, how about nymph?? Please correct.
Lines 356-358 (page 9)
Please rephrase the sentence; it sounded ‘funny’ (i.e. was not worded correctly).
Line 366 (page 9)
Delete ‘sublethal concentration (median lethal concentration)’ – this is a redundancy of what you said in line 363. Simply say LC50 is sufficient.
Line 402 (page 9)
It quantified (with “d”, not “s”).
Comments on the Quality of English LanguageSee comments above
Reviewer 2 Report
Comments and Suggestions for Authors
This manuscript is a review of the advancements in utilizing life history tables for pest management. The authors work to identify multiple examples from different fields of entomological study to highlight how life history tables may be used. The authors also claim that age-stage, two-sex life history tables are more effective than the older one-sex tables.
The examples the authors use in each sub-section to demonstrate how life history tables are used are suitable. The authors highlight the information that can be obtained from life history tables, and then how that may impact control efforts for a range of entomological pests. Some of the paragraphs seem repetitive, however, as the authors frequently repeat phrases related to the details that life history tables can provide (information on survivorship, mortality rates, rate of increase, etc.). Especially towards the earlier paragraphs these are repeated practically for each section. I encourage the authors to revise the manuscript such that there is a sub-section early on about the data that can be obtained from life history tables and how this data can be used (which is currently partially highlighted in the introduction. The authors can then limit each of the sub-sections with examples to highlight and explain the examples – not needing to explain each time that they are measuring fecundity, mortality, etc.
The authors early of suggest that two-sex life history tables are a revolutionary leap in use collecting data. However, there is not a strong case made that two-sex tables provide critical data that is missing from one-sex tables. Nor is there any details about the comparative difficulty of collecting data for two-sex life history tables. In the manuscript, the authors highlight only one example of when a two-sex study was implemented, nor do they indicate what additional details or information may be obtained relative to the other studies highlighted. As such, there is limited support for their title “advancements in age-stage, two-sex life tables”.
I encourage the author to identify additional examples of using two-sex life tables for integrated pest management, or more clearly highlight the benefit that two-stage life tables would provide in the examples of using life history tables. Otherwise, their title is misleading.
Lastly, early in the manuscript the authors highlight the need for and importance of using these life history tables in control efforts, which by their own statement, is not done frequently. However, the authors make no effort to identify ways that this can be rectified, either through communication, education, partnerships, or some other method. I suggest the author address this potential issue.
As a population biologist, I understand the importance that life history tables can have in understanding population abundance. The authors highlight this to a degree but fail to explain how to encourage their use in more real-world situations. Their application to IPM strategies is currently limited, and there no suggestion as to how to improve this.
Reviewer 3 Report
Comments and Suggestions for Authors
The manuscript provides a comprehensive review of the advantages of using age-stage, two-sex life tables in various aspects of Integrated Pest Management (IPM). The study effectively highlights the broad applications of life tables in controlling agricultural pests through natural enemies (predators and parasitoids), biological agents (such as plant derivatives), and other non-chemical and chemical methods. Additionally, the authors discuss the role of life tables in managing vector insect populations and invasive species. The findings underscore the necessity of understanding insect life cycles to enhance pest control strategies within IPM frameworks.
Overall, the manuscript is well-written and makes a valuable contribution to the field. However, some areas require improvement before acceptance. A major revision is suggested to address the following concerns:
- The study would benefit from additional examples supporting the practical application of life tables in pest management. More case studies or experimental results would strengthen the review.
- A dedicated discussion section should be added to provide a deeper analysis of the reviewed literature and highlight existing gaps.
- This future directions section currently lacks sufficient references and reads as an opinion piece. The authors should include relevant citations to support their claims and provide a more structured outlook.
- The figures require more detailed descriptions to enhance clarity. The captions should explicitly explain the significance and interpretation of the presented data.
- All scientific names should include the author’s name upon first mention.
- Taxonomic classification (order and family) should be provided for each species discussed in the manuscript.
- A thorough grammar check is necessary to improve sentence structure and clarity. Some sentences should be revised for conciseness and readability.
With this revision, the manuscript will be well-suited for publication.

Round 2
Reviewer 1 Report
Comments and Suggestions for Authors
A brief summary
The authors of this manuscript conducted extensive literature reviews on the advancement of life tables with the hope to integrate two-sex life tables in different technologies as part of IPM. As was in the first version of this manuscript, it is organized well. The figures helped visualizing what the authors attempted to convey; after revision, the figures were much improved with more detailed information on the legends. The authors have done an excellent job in improving the English language which has improved tremendously as well, much easier to follow and read. Only minor updates are needed as suggested below.
Specific comments (page numbering is based on the tracking file):
Throughout the document, ‘integrated pest management’ was mentioned. Since it was already abbreviated in the Simple Summary as IPM, it can be simplified by abbreviating throughout the manuscript.
Title (pg. 1)
Lines 2-3
I previously suggested to remove the ‘two-sex’ in the title but after reading this revised review manuscript, it appears that the majority of your discussion is still around two-sex life tables. May I suggest a revised title to “Advancements in life tables applied to integrated pest management with an emphasis on two-sex life tables”?
Abstract (pg.1)
Line 38-41
The sentences here were confusing or unclear. Please rewrite.
Subtitle 2 (pg. 3)
Line 110
“… in pest biological control” --> Change to “… in biological control of pests”
Line 136
Change ‘increasing’ to ‘increased’
Pg. 4
Lines 144-147
The sentence is confusing, please rewrite.
Pg. 5
Line 196
Replace ‘with a high efficiency’ with ‘efficiently’.
Pg. 6
Line 282
Move the word ‘concentrations’ before ‘(LC50)’ --> Lethal concentrations (LC50)
Pg. 7
Line 311
Italicize Helicoverpa armigera
Pg. 8
Lines 374-378
Sentence is too long and confusing. Split in two sentence and clarify.
Subtitle 3 (pg. 14)
Line 586
“… in pest chemical control” --> Reword to “… in chemical control of pests”
Subtitle 4 (pg. 19)
Line 753
Italicize Trichogramma
Reviewer 2 Report
Comments and Suggestions for Authors
I appreciate the revisions made and feel my concerns have been addressed sufficiently.
Author Response
We would like to express our sincere gratitude for your thorough review and constructive comments on our manuscript.
Reviewer 3 Report
Comments and Suggestions for Authors
Because of all the track changes and corrections, it's a bit harder to follow the text, but here are my comments:
General:
- The authors adopted all comments and improved their manuscript, which is now ready for publication.
- Before that, there are still many minor corrections that need to be made, mostly concerning typos and inconsistencies.
Simple Summary
- Lines 16-26: is correct
Abstract
- Lines 27-41: I see that English is improved
Keywords
- Lines 42-43: uniformed lowercase
1. Introduction
- Lines 46-66: This part was correctly written in the previous version of the manuscript
- Lines 47-93: Authors rewritten this part and expanded it.
- Lines 94-109: several errors have been corrected, and reference numbering has also been applied
- Application of life tables in pest biological control
- 1 Application in agricultural pest control
- Lines 112-122: The text has been rewritten and is now much better. The authors also referred to Table S1. No further comments.
- 1. 1 Evaluation of the pest control capacity of natural enemies of pests
- Lines 134-137; 140-142: This part has been rewritten; repetitive information has been deleted.
- Lines 147-148: The sentence is supplemented: ‘Natural enemies are usually classified as predators, parasitoids, and pathogens.’
- Lines 150-163: Paragraph corrected.
- Lines 164-205: The paragraph has been corrected, rewritten and rephrased. Authors of species descriptions were added for every species that appears in the main body for the first time.
- Lines 206-256: Whole this paragraph has been rewritten. Authors of the species were added. No comments
- Lines 257-289: This paragraph has also been rewritten. English has improved.
- Line 269: As you did for Plutella xylostella (Line 243) or Tenebrio molitor L. (Line 247) wherever it says Linnaeus abbreviate to L. as it is for example (Coccinella septempunctata Linnaeus). Search and replace.
- Line 275: delete the article (a) in front of Culex
- Line 277: Is this correctly written (22. 6 h)?
- Line 283: correct Pipiens to pipiens
- 1. 2 Evaluation of the pest control capacity of bio-pesticides
- Lines 291-317: The paragraph is rewritten.
- Line 291: for Helicoverpa armigera give the author and italize
- Line 312: Habrobracon hebetor (Say) or Bracon hebetor Say as it was described by Say.
- 1. 3 Screening of insect-resistant plant species
- Lines 349-382: The paragraph is rewritten.
- Line 379: Toxoptera aurantii give the author (Please check all the species and add the author’s name)
- 2 Application in vector insect control
- Lines 393-429: The paragraph is rewritten. No comments.
- 2. 1 Assessing the risk of transmission by vector insects
- Lines 430-456: This paragraph is almost completely rewritten. Like for Toxoplasma, add author for Plasmodium knowlesi (Line 446).
- 2 Guiding the selection of insecticides
- Lines 627-655: the paragraph has been completely rewritten
- Line 647: Bradysia difformis to difformis
- 3 Guiding the application of insecticides
Lines 703-740: the paragraph has been rewritten, no comments
- Discussion and future perspectives
This part is rewritten and slightly shortened compared to the original text
References
Now the order/numbers is correct
